# Mechanisms and contextual factors related to key elements of a successful integrated community-based approach aimed at reducing socioeconomic health inequalities in the Netherlands: A realist evaluation perspective

Lisa Wilderink[1,2]*, Annemijn Visscher[3], Ingrid Bakker[2], Albertine J. Schuit[4], Jacob C. Seidell[1], Carry M. Renders[1]

1 Department of Health Sciences, Faculty of Sciences, Amsterdam Public Health Research Institute, Vrije Universiteit Amsterdam, Amsterdam, The Netherlands, 2 Department of Healthy Society, Windesheim University of Applied Sciences, Zwolle, The Netherlands, 3 Research Department of the Municipal Public Health Service Organization Flevoland, Lelystad, The Netherlands, 4 School of Social and Behavioral Sciences, Tilburg University, Tilburg, The Netherlands

* l.wilderink@vu.nl

**Data Availability Statement:** All qualitative data files are available from the Data Archiving and

## Abstract

### Background

Reducing health inequalities is a challenge for policymakers and civil society. A multisectoral and multilevel approach is most promising to reduce those inequalities. Previous research showed what key elements of Zwolle Healthy City, an integrated community-based approach aimed at reducing socioeconomic health inequalities, are. To fully understand approaches that are complex and context dependent, questions as 'how does the intervention work' and 'in what context does it work' are just as important as 'what works'. The current study aimed to identify mechanisms and contextual factors associated with the key elements of Zwolle Healthy City, using a realist evaluation perspective.

### Methods

Transcripts of semi-structured interviews with a wide range of local professionals were used (n = 29). Following realist evaluation logic in the analysis of this primary data, context-mechanism-outcome configurations were identified and thereafter discussed with experts (n = 5).

### Results

How mechanisms (M) in certain contexts (C) were of influence on the key elements (O) of the Zwolle Healthy City approach are described. For example, how, in the context of the responsible aldermen embracing the approach (C), regular meetings with the aldermen (M) increased support for the approach among involved professionals (O). Or, how, in the

Networking Services (DANS) repository database and can be found via https://doi.org/10.17026/dans-zkf-46xc.

**Funding:** This project was funded by the Netherlands Organization for Health Research and Development (ZonMw), Den Haag, The Netherlands, under Grant number 531001314. The funding body was not involved in the design of the study, the collection, analysis and interpretation of the data, nor in writing the manuscript. Funder website: https://www.zonmw.nl/nl/.

**Competing interests:** The authors have declared that no competing interests exist.

context of available financial resources (C), assigning a program manager (M) contributed positively to coordination and communication (O). All 36 context-mechanism-outcome configurations can be found in the repository.

## Conclusion

This study showed what mechanisms and contextual factors are associated with the key elements of Zwolle Healthy City. By applying realist evaluation logic in the analysis of primary qualitative data we were able to disentangle the complexity of processes of this whole system approach and show this complexity in a structured manner. Also, by describing the context in which the Zwolle Healthy City approach is implemented, we contribute to the transferability of this approach across different contexts.

## Background

Reducing socioeconomic health inequalities is a challenge for policymakers and civil society. All over the world, people with a relatively low socioeconomic position (SEP) in their society have a shorter healthy life expectancy compared to people with a high SEP [1]. Socio-economic health inequalities are mainly caused by an unequal distribution of individual and environmental determinants of health related behaviors [2–4]. Determinants on the individual level are, e.g. psychological distress, lower self-efficacy for behavioral change, a lack of knowledge about healthy choices and a lower level of health literacy [5–7]. But also unfavorable circumstances in the social and physical environment are of influence, as the environment in underprivileged neighborhoods offers less support and fewer opportunities for healthy behavior [8–11]. Hence, the complexity of health inequalities is that they are caused by circumstances at different levels (e.g. the individual, social and community, and socioeconomic, cultural and environmental level) and circumstances in different sectors (e.g. work, health care, housing). For that reason, a coherent multisectoral and multilevel approach aimed at changing those circumstances is most promising to reduce socioeconomic health inequalities [12]. It is important to actively involve community members in such an approach, because they are at heart of the complex system [13].

A complex whole system approach to tackle socioeconomic health inequalities requires an evaluation that addresses those complexities. Recent literature has stressed the importance of the interaction between determinants on different levels in complex system thinking and has acknowledged that these relationships may not be linear and can be bidirectional [14–18]. For example, food choices are strongly determined by the supply of foods and drinks but that, in turn, is affected by demands of customers.

Moreover, the complexity of evaluating whole system approaches lies not only in the complex relations between determinants from different levels, but also that these relations may be context specific. We should not consider public health approaches aimed at reducing socioeconomic health inequalities as "one size fits all" but consider them as context dependent, because the effectiveness of these approaches is critically influenced by their implementation at a specific time in a given context [19–21]. However, until now, most studies that have evaluated complex approaches report limited information on contextual factors [22]. To fully understand approaches that are complex and context dependent, questions as 'how does the intervention work' and 'in what context does it work' are just as important as 'what works'.

In recent years, many municipalities in the Netherlands have implemented integrated public health approaches and evaluated its process and effect [23–26]. However, studies that

answer questions such as 'how does the intervention work" and 'in what context' are limited [27, 28]. Since 2010, the municipality of Zwolle, a Dutch city, has implemented a multilevel and multisectoral approach aimed at reducing socioeconomic health inequalities. Many organizations from different sectors that can potentially promote healthy behavior have been involved, for example healthcare- and welfare organizations, municipal health and sports services and the local government. This local approach, called 'Zwolle Healthy City', made Zwolle the first Dutch JOGG (Dutch acronym for 'Youth at a Healthy Weight') municipality. The JOGG approach is an integrated community-based approach that includes a variety of interventions that target health behavior determinants from the individual and environmental level, coordinated locally by a program manager [26, 29]. Today, more than 50 percent of all municipalities in the Netherlands have implemented the JOGG approach [30].

The Zwolle Healthy City approach has been proven successful with respect to stimulating healthy behavior and reducing socioeconomic health inequalities: the prevalence of overweight in children in the two neighborhoods with the lowest SEP decreased more in comparison to other neighborhoods in Zwolle [31]. Previous research identified key elements of this successful approach [32].

For the evaluation of this multilevel and multisector approach in complex whole system thinking it is not only important to describe what the key elements of an approach are, but also which dynamic processes can reinforce the impact of the key elements (i.e. positive feedback loops operating in the system) and what contextual factors are of influence. The aim of the current study is therefore to identify mechanisms and contextual factors associated with the key elements of Zwolle Healthy City, by using a *realist evaluation* perspective. This perspective will be used in the data analysis of primary qualitative data, collected prior to this study with interviews with various stakeholders to identify key elements of the approach, to better understand what mechanisms can be stimulated that influence the key elements and how to adapt the local approach to other contexts. Realist evaluation is suitable for this aim because realism is based on the idea that the same intervention, or approach in this case, will not work the same for everyone everywhere. We aim to unravel what specifically in the Zwolle Healthy City approach contributed to a successful approach, in order to show what mechanisms and contextual factors should not only be strengthened by the Zwolle Healthy City approach, but also pursued by other approaches.

The initial programme theory that underpins the Zwolle Healthy City approach is inspired by the EPODE approach developed in France and was described as a logic model in previous research [32]. After implementing the approach in Zwolle, the approach is widespread throughout Dutch municipalities as the JOGG approach. This current study is using a realist evaluation perspective to refine and add detail to this initial programme theory with related mechanisms and contextual factors by using stakeholder perspectives.

## Methods

### Approach and design

To identify mechanisms and contextual factors that are associated with the key elements of the Zwolle Healthy City approach, this study applied *realist evaluation* logic in a case study design. Using a realist evaluation perspective provided the opportunity to answer questions as 'how does it work?' and 'under which conditions does it work?', rather than just answering the question 'does it work?' [33]. In realist evaluation it is acknowledged that mechanisms (M) of an approach may only work in a certain context (C), and that the combination of both can lead to a certain outcome (O) [34, 35]. In this study, transcripts of semi-structured interviews with a wide range of local professionals were used to identify so-called context-mechanism-outcome

configurations (C + M = O). These interviews were undertaken prior to the design of this realist evaluation study with the aim to identify key elements. Semi-structured interviews were carried out with professionals from different organizations and backgrounds into their experiences with the implementation and execution of the approach during the period they were involved (during the entire period of 2010–2018, or part thereof)'. After publishing on those insights [32], we concluded that the data was rich enough to identify mechanisms and contextual factors that led to the key elements. The latter are seen as outcomes of the CMO configurations in this study. The analysis of this secondary data was therefore suitable for deepening the initial programme theory (the logic model of the Zwolle Healthy City approach) with mechanisms and contextual factors related to the key elements.

The study protocol was approved by the Medical Ethics Review Committee (METc) of VU University Medical Center, registered with the US Office for Human Research Protections (OHRP) as IRB00002991. The METc confirmed the study did not require medical ethical approval under Dutch legislation on medical trials (2018.601). All procedures performed were in accordance with the WMA declaration of Helsinki.

## Programme theory

Realist evaluation combines programme theory with stakeholders' theory [33–35]. The view and perceptions of stakeholders in practice on how the programme works can help to refine and enrich the initial programme theory. The initial programme theory that underpins the Zwolle Healthy City approach, and was inspired by the French EPODE approach, was described as a logic model in previous research [32]. This approach is nowadays implemented in more than 50 percent of Dutch municipalities. It states that through political commitment, public-private partnership, social marketing and combining prevention and care in a local coordinated approach that is monitored and evaluated, healthy behavior is stimulated. This current study is using a realist evaluation perspective to refine and add detail to this initial programme theory with related mechanisms and contextual factors by using stakeholder perspectives. These stakeholders' perspectives can help us answering questions as 'how does it work?' and 'under which conditions does it work?' and with that add more detail to the initial programme theory.

## Study population

In the previous study where the secondary data were collected, respondents were purposively sampled to create a heterogeneous interview sample. All respondents (n = 29) were involved with the implementation of the Zwolle Healthy City approach somewhere in the period between 2010 and 2018. Respondents worked at the strategic (n = 4), tactical (n = 17) or operational level (n = 8) of six involved organizations (see Table 1). At the strategic level, four

Table 1. Overview of involved organizations and the number of respondents that work for that organization (*n* = 29).

|  | Strategic | Tactical | Operational | Total |
|---|---|---|---|---|
| Local government | 4 | 2 | 0 | 6 |
| University of applied sciences | 0 | 4 | 0 | 4 |
| Municipal public health service | 0 | 2 | 0 | 2 |
| Municipal sports service | 0 | 2 | 3 | 5 |
| Welfare organization | 0 | 4 | 5 | 9 |
| Home care organization | 0 | 3 | 0 | 3 |
| **Total** | 4 | 17 | 8 | **29** |

municipal executive councilors who had been involved in the approach since the start in 2010 participated individually in an interview. At the tactical level respondents were involved as a manager, policy advisor or (associate) professor. Professionals at the operational level were involved as sports club advisor, community sports worker, child worker, youth worker or community worker.

## Data collection

The secondary data used in this study were originally collected by the first author (LW) in 2019. Semi-structured interviews were held and analyzed with the aim to identify key elements of the Zwolle Healthy City approach. The interview guide consisted of open-ended and follow-up questions and contained the following topics: local structure and coordination, support, interventions and activities, aligning prevention and healthcare, monitoring and evaluation, collaboration, and citizen participation. More information on where the interview guide was drawn upon and how the data was collected in 2019 can be found elsewhere [32].

**Expert meeting.** In this current study the CMO configurations that followed after data analysis were discussed with experts (n = 5) to verify whether the identified mechanisms and contextual factors are recognizable and whether important factors were missing. The group of experts was purposively sampled and consisted of a strategic advisor integrated health policy at the knowledge institute RIVM (the National Institute for Public Health and the Environment), a strategic advisor at Pharos (expertise center for health inequalities), a senior advisor at JOGG, a policy advisor at the local government and an advisor public health at the municipal public health service. The consulted experts were asked to reflect on the identified mechanisms and contextual factors that were extracted from the data analysis of the semi-structured interviews. The experts generally agreed with the preliminary results and their additional comments are described in a separate paragraph in the results section of this paper.

## Data analysis

For this study all 29 available transcripts (secondary data) were again coded independently by two researchers (LW and AV). A realist perspective was now used in the data analysis by labeling relevant quotes as contextual factor (C) or mechanism (M) if present and mentioned by the respondent in combination with one of the nine key elements [34, 36]. Key elements were labelled as outcomes (O). See Table 2 for an overview of definitions and operational descriptions of contextual factors, mechanisms, and outcomes. By actively exploring mentioned combinations of contextual factors and mechanisms by the respondents that led to the outcomes, the realist evaluation principle of *generative causation* was pursued [33]. The principle of *retroduction* can be seen as an iterative inductive and deductive cycle and enables us to find patters that answer the question '*why* do things appear as they do?'. In the qualitative data phrases were highlighted that include these relationships, for example "Because we initiated this. . . this happened", or, "Because these circumstances were there. . . this was made easier".

Following these realist principles all combinations of the three (C, M and O) were listed as identified context-mechanism-outcome configurations, as linked sets [34]. Thereafter the list of 117 CMO configurations was deduced to 36, in discussion with members of the research team (LW, IB, CR). Aim of those discussions was to (a) combine detailed CMO configurations into more general CMO configurations, (b) determine the character of a factor as C, M or O when it was multi interpretable by using the definitions described in Table 2 and finally, (c) assign the mechanisms and contextual factors to the one outcome that, in our opinion, was the best fit. This was discussed until consensus was reached.

**Table 2. Definitions, operational description and elaboration of 'context', 'mechanism' and 'outcome'.**

|  | Definition | Operational description | Elaboration |
|---|---|---|---|
| **Context** | For who and in what circumstances does the mechanism work. A relationship between mechanisms and their effects is not fixed but contingent. | Something happening outside control of the Zwolle Healthy City approach, but of influence on the outcome. | A contextual factor can be • Situational: in a specific moment in time Or • Continuous: ongoing And • Supportive to the mechanism Or • Restraining to the mechanism E.g. policy, (financial) resources, organizational factors |
| **Mechanism** | Responsible for the relationship between context and outcome. The cause of change. | The response that actors have to activities or actions in the Zwolle Healthy City approach that, in the context, contribute to the outcome. | Activities and actions taken by professionals on the • strategic level • tactic level • operational level of involved organizations. |
| **Outcome** | Result of the combination of a mechanism with one or more supportive or restraining contextual factors. | Key elements of Zwolle Healthy City. | Nine key elements that were previously described by Wilderink et al. (2020) [32]. |

Based on Byng et al., 2005; Herens et al., 2017; Pawson, 2002; Pawson & Tilley, 1997

## Results

The analysis of transcripts of 29 semi-structured interviews led to 36 context-mechanism-outcome (CMO) configurations, describing *how* the combination of mechanisms and contextual factors led to outcome of the Zwolle Healthy City approach. A full description of all 36 CMO configurations can be found in the repository [see S1 File], including configurations at strategical, tactical and operational level. In S2 File, an infographic visually shows all CMO configuration. In the following paragraphs we describe 22 CMO configurations specifically related to the 9 key elements of Zwolle Healthy City (outcomes, O), and in the closing paragraph 3 contextual factors that have an overall influence. We chose to describe mainly configurations at the tactical level in the result section of this paper, because this level takes a central position in the Zwolle Healthy City approach. The following results show how (with what mechanisms) and in what context the key elements of Zwolle Healthy City were implemented, and with that, add more detail to initial programme theory.

### Outcome 1: Collaboration between a variety of local organizations

Formulating goals for the Zwolle Healthy City approach collectively with a group of professionals at the tactical level of involved organizations (M) when there were good relationships between these professionals (C) contributed positively to collaboration (CMO configuration 1.1). Also jointly identifying roles and responsibilities of each involved network organization (M), contributed positively to collaboration, when these roles and responsibilities align with goals and structures of the individual organizations (C) (CMO configuration 1.2). Both mechanisms in these contexts provided clarity in the responsibilities of organizations in the implementation process, made involved professionals aware of what their tasks are and increased involved professionals' feeling of being part of the approach. In the context of local existing structures in the Netherlands in which local organizations function and depend on each other (C), coming together as this group of professionals at the tactical level of involved

organizations in regularly monthly meetings (M) contributed positive to collaboration. This was the case because in those meetings the implementation of the approach was coordinated, collaborations were formed, and knowledge exchanged. This also let to more clarity among involved professionals about their specific tasks and made them see the advantage of collaboration (CMO configuration 1.3). Moreover, the fact that these meetings were chaired by a program manager who coordinated the network (M), with financial resources for this task (C), made a positive contribution to collaboration (CMO configuration 1.4).

## Outcome 2: Support on three levels: Strategic, tactical and operational

In the context of the responsible aldermen embracing the approach (C), organizing regular meetings (e.g. twice a year) between the network of involved organizations and the aldermen (M) increased support on the tactical level (CMO configuration 2.1). Through those meetings, once or twice a year, involved professionals felt like they could influence policy measures made by the alderman. A second combination of a mechanism and contextual factor that are positively related to this outcome was assigning professionals that functioned as a bridge between the tactical level (managers) and operational level (M), with the contextual factor that there are financial resources available to assign these professionals (C) (CMO configuration 2.2). Those professionals were working for the municipal health services and had the task to convert the policy into neighborhood-specific practices, and the other way around. As a consequence, the vision of Zwolle Healthy City was successfully translated into a realistic and broadly endorsed activity program, which contributed to its support at the operational level because his led to clarity among professionals working at this level about their tasks and roles.

Aligning the approach with the daily work of professionals (M) also increased the support at the operational level, which was easier when the work agreements between local government and implementing organizations are in line with the approach (C) (CMO configuration 2.3). In this way, professionals of involved organizations did not feel obligated to do extra activities for Zwolle Healthy City, which increased their support.

## Outcome 3: Coordination and communication

Assigning a program manager who chaired the network (M), with financial resources available for this function (C), contributed positively to the outcome of coordination and communication, because this led to more clarity among involved professionals about their specific tasks (CMO configuration 3.1). The program manager was responsible for communicating about the meetings at the tactical level and coordinating the implementation and assignments of tasks for each organization. For example, the monitoring and evaluation of the approach was coordinated by the program manager and the involved professor of the University of Applied Sciences. Assigning professionals that formed a bridge between professionals working at the tactical level and those working at the operational level of the approach (M), also with financial resources available for these functions (C), contributed positively to this third outcome of coordination and communication as well: they were responsible for translating policy to implementation at the operational level and vice versa (CMO configuration 3.2).

Additionally, assigning an operational professional as an internal coordinator for the approach in every involved organization to coordinate the implementation for the approach within their own organization (M), with financial resources available for these functions (C), contributed positively to coordination and communication (CMO configuration 3.3). This was the case because those internal coordinators functioned as internal ambassadors for the approach and were responsible for generating attention for the Zwolle Healthy City approach among their colleagues, the coordination of the implementation, and embedding the approach

in their organization. Because of this, professionals felt more urge and support to work on the goals of Zwolle Healthy City.

## Outcome 4: Embeddedness of the approach in organization's policy and processes

Integrating the theme of the Zwolle Healthy City approach, 'healthy lifestyle', in regular activities of involved organizations (M) facilitated embeddedness of the approach in the organization's processes (CMO configuration 4.1), because this made that professionals did not feel they have to do something in addition to their regular work. However, this is not the case in the context when the theme 'healthy lifestyle' was not seen as a core responsibility of involved organizations (C). When the goals of Zwolle Healthy City were seen as too divergent from the goals of the involved organizations, the embeddedness was difficult.

Including the theme 'healthy lifestyle' in the policy of organizations and in formal agreements between the local government (financing organization) and executive organizations (M), if local policy aims to promote a healthy lifestyle (C), contributed positively to embeddedness of the approach (CMO configuration 4.2). In this way, organizations and professionals working for these organizations were obligated to work on the theme 'healthy lifestyle' irrespective of the motivation of the person responsible for the implementation.

## Outcome 5: Collaboration with private organizations

Because differences in culture, language and interest between public and private organizations exist (C), exploring opportunities to work together with potential private organizations with these private organizations themselves (M) contributed positively to collaboration (CMO configuration 5.1). This increased not only their motivation to contribute, but also through this jointly exploring, private organizations came up with ideas themselves that were in line with both their mission and goals, and the mission and goals of the integrated approach.

Specifically collaborating with *local* private organizations (M) that felt involved in their neighborhood and recognized the importance of promoting a healthy lifestyle for their customers and employees (C) contributed positively to outcome 5 (CMO configuration 5.3). This worked well because the local organizations could, in contrast to national organizations, play a specific role in the implementation of the approach. A local supermarket for example was able to provide free fruit for the participating children of an organized sports tournament.

## Outcome 6: Collaboration with citizens

In the Zwolle Healthy City approach, there were several ways to stimulate collaboration between professionals and citizens. For example by giving citizens responsibilities in the organization of activities (M), together with professionals who had time for coordination (C), which increased the willingness of citizens to participate and with that contributed positively to this outcome (CMO configuration 6.1). Involving citizens in a project or activity right from the start (M), in the context where welfare organizations who had this responsibility are well organized in the municipality (C) contributed positively (CMO configuration 6.2). Investing in trust by taking time to listen to citizens' opinions and include these in further development and implementation (M), especially in a context where citizens consider trust important (C), contributed positively as well to this outcome (CMO configuration 6.3). The combination of both these mechanisms and contextual factors prevented that citizens feel like things are already decided upon which can make them feel unimportant. Moreover, using existing social infrastructure in the neighborhood (M), in a context where people with a low SEP are often hard to reach and involve (C), stimulated collaboration as well, because this made it easier for

professionals to reach them (CMO configuration 6.4). For example existing groups or ambassadors were used to facilitate getting in contact with citizens who are difficult to reach. Ambassadors are socially engaged individuals with access to both informal and formal networks and are therefore suitable for making the first contact between persons who don't meet usually. More precisely, a single mother who takes an active role in the parent council at school, can through her network more easily approach other single mothers in the neighborhood.

### Outcome 7: Profiling the approach like a brand

Linking the successes of involved organizations to the Zwolle Healthy City approach (M) contributed positively to profiling the approach (CMO configuration 7.1). Through this, people and organizations became familiar with the approach and saw the added value of contributing. This led to new collaboration opportunities and support for the approach. A restraining contextual factor for this mechanism was that the involved organizations saw the approach as embedded in their own organization (C), and for that reason, preferred to communicate as if the successes came from them, and not from the approach.

### Outcome 8: Move along with, and take advantage of, (local and national) opportunities

Making use of national funding opportunities by involved organizations of Zwolle Healthy City (M), in a context where there is national attention for a healthy lifestyle, contributed positively to this outcome (for example a national funding for healthy schools) (CMO configuration 8.1). By doing this, there were financial resources to continue local activities and interventions.

### Outcome 9: Continuous monitoring and evaluation goals and process, and learning from the results

Making results and conclusions clear and communicating about this towards other involved organizations within the network (M) was a mechanism that contributed positively to monitoring and evaluation (CMO configuration 9.1) This was possible with the supportive contextual factor of having a University (of Applied Sciences) situated in the municipality, responsible for monitoring and evaluation (C). Through making results and conclusions clear, involved professionals saw what participating in monitoring and evaluation can yield. If researchers showed the results and conclusions, involved organizations that implemented the approach saw the added value of it, which made them more willing to contribute. Drawing conclusions from monitoring and evaluation to adapt the approach based on those conclusions (M) was an important mechanism for learning from the results (CMO configuration 9.2), and this increased the support for monitoring and evaluation as well because professionals saw that it actually helps in further developing the approach. However, this mechanism was restrained by the contextual factor where involved organizations do not hold each other accountable for the results (C). There is no reckoning between involved organization, which works restraining for making adaptations in the implementation based on negative evaluations.

### Overall contextual factors

A first overall contextual factor that can work restraining (or supportive) on the above described outcomes was the difference (or similarities) in culture, language or interest that different organizations that worked together in the integrated community-based approach faced.

For example public and private organizations often have different working methods: where private organizations tend to think more often in terms of action, public organizations need in general more time to develop ideas and generate support. Also differences between researchers and professionals in daily practice played a role, as professionals did not always see what monitoring and evaluation could yield.

A second overall contextual factor was the availability of financial resources. This was necessary for example to appoint a program manager, or professionals who formed a bridge between the tactical and operational level, or the internal coordinator at the operational level who functioned as an ambassador within their own organization, or the operational professionals who had the task to facilitate collaboration with citizens. They all had to spend hours on their (sometimes additional) tasks, for which financial resources were required but not always available.

A third overall contextual factor concerned relations. Good relations between the persons that represented involved network organizations and good relations with potential collaborating private organizations. This made mechanisms that lead to successful collaboration and support for the approach possible.

## Expert meeting results

Above described mechanisms and contextual factors that were of influence of the nine key elements (outcomes) were verified in an expert meeting. In general, the consulted experts recognized the results and agreed on the preliminary identified CMO configurations. Minor adjustments in the text were made, concerning structure and language. There were three additional comments.

First, the consulted experts advised to combine some mechanisms, as their function in the system is the same. This advice led to combining the mechanisms 'making visible to colleagues what is happening within the approach' and 'share successes and celebrate together' that contributed to outcome 2 (support on three levels: strategic, tactical and operational) into the mechanism 'extensive internal and external communication and celebrating successes together' (CMO configuration 2.6). Likewise, the mechanisms 'using existing groups' and 'using ambassadors in the neighborhood' that contribute to outcome 6 (collaboration with citizens) were combined into the mechanism 'using existing social infrastructure in the neighborhood as existing groups and ambassadors' (CMO configuration 6.4).

Second, according to the experts, the described overall contextual factors should not be assigned as 'supportive' or 'restraining' (what they were before), as they can function as both. For example, on the one hand, differences in culture, language and interest between public and private organization can be seen as restraining, as it takes more effort to be 'on the same page' when collaborating. On the other hand, the experts mentioned that those differences can work supportive as well, since they bring an alternative perspective and expertise that can be helpful in the implementation of the approach.

Third, the consulted experts identified two contextual factors that were previously missed in the results: a mandate from the strategic level of involved organizations for professionals at the tactical and operational level of involved organizations to work on the approach, and secondly, changing organizational structures on which embedding was based. According to the experts, a contextual factor is that professionals working at the tactical and operational level of involved organizations have decision-making authority and the mandate from the strategic level that they are allowed to work on the implementation of the approach. Moreover, a restraining contextual factor, according to the experts, is that the embedding of the approach was based on organizational structures of that specific moment in time. When those structures

changed or disappeared, the embedding faded. Those contextual factors were added to the repository.

## Discussion

In this study a realist perspective was used in the analysis of secondary qualitative data to identify mechanisms and contextual factors associated with key elements of Zwolle Healthy City, an integrated community-based approach aimed at reducing socioeconomic health inequalities. We displayed 36 CMO configurations (Context + Mechanism = Outcome) in the repository and described mechanisms and contextual factors at the tactical level in the results section of this paper, because this level takes a central position in the Zwolle Healthy City approach.

### Reflections on initial programme theory

Realist evaluation set out to develop, support, refute or refine aspects of initial programme theory. In our realist evaluation study, the aim was to add detail to the programme theory of Zwolle Healthy City, captured in the logic model of the approach. The logic model describes that through political commitment, public-private partnership, social marketing and combining prevention and care in a local coordinated approach that is monitored and evaluated, healthy behavior is stimulated [32]. Our study showed how (with what mechanisms) and in what context these aspects were implemented, and with that, based on stakeholder perspectives, enriched this initial programme theory. Among other things, for example it is specified that for this local coordination, the role of the programme manager takes a central place (in a context where there are financial resources available for this coordinating function) and the role of internal coordinators at the operational level is also of great value. Moreover, in our results we describe how collaboration with citizens can be improved, which goes beyond social marketing techniques as were described in the initial programme theory. These refinements and details added to the initial programme theory is described in the results section and repository.

### Findings in relation to current literature

This study answers to the call of Sniehotta et al. (2017) [37] in the Lancet which states that researchers should study the interaction of multiple levels within systems relevant to population health and health inequalities. According to Sniehotta and colleagues, the association between individual and environment in system approaches is continuous, dynamic, and relational [37]. For that reason, identifying this dynamic interaction in the system requires "systematic modelling of the mechanisms and effects in the wider physical, economic, policy and sociocultural environments that interact with individual and behavioral factors". By identifying mechanisms and contextual factors in the multilevel and multisectoral Zwolle Healthy City approach, we responded to this call. In the approach of Zwolle Healthy City the complex mix of determinants that influences socio economic health inequalities are recognized and the synergism between individual and environmental determinants is acknowledged [38, 39]. By using a realist evaluation perspective we were able to disentangle the complexity of processes of this whole system approach and show this complexity in a structured manner. Moreover, the scientific literature highlights the complexities involved in transferring local population-level public health policy interventions to other contexts [40]. Therefore, multiple authors advocate for taking context into account when evaluation public health interventions [19, 20]. By describing the context in which the Zwolle Healthy City approach is implemented, we contribute to the transferability of this approach across different contexts.

The mechanisms and contextual factors identified in this current study are mostly in line with recent results of literature that evaluated complex public health approaches using different methods. Studies on healthy behavior approaches for example show the importance of availability of 'preconditions' as contextual factors: e.g. the capacity to implement in terms of (financial) resources and time [41, 42] or the mandate from the strategical level of involved organizations [43]. Also the mechanisms related to coordination and communication are in line with literature: coordination is needed from the strategic, tactical and operational level and leadership (an "ambassador" or "linking pin") at every level is desirable. In a previous study on the role of evidence, context, and facilitation in implementation studies this was described as communication within and across teams and departments [42]. Assigning leaders at all levels identified in this study as a mechanism (for example a program manager and internal coordinators in involved organizations) is also found an important factor in multiple other recent studies on whole system approaches [44–47].

A precondition that is an important enabler of implementation plans according to previous research is understanding of the policy context by professionals who are responsible for implementation [48]. This is in line with our finding of the importance of a 'bridge' between the tactical and operational level, formed by professionals who were responsible for translating policy to implementation at the operational level. The overall contextual factor of good relationships for successful multisectoral collaboration is recognized in a study on collaboration between the public health sector and other policy sectors in sixteen Dutch municipalities [49].

The outcome of collaborating with citizens in integrated public health approaches is broadly studied [50, 51]. In contrast to previous research, mechanisms or contextual factors related to ethics as expectations management, issues of power and group dynamics are not found in our study [52–55].

The importance of adapting the approach based on conclusions of monitoring and evaluation is recognized in previous studies and often described as systematic assessing processes and/or outcomes of a program with the aim to further develop and improve in a learning cycle [56, 57]. Evaluating impact is also mentioned in previous research [58], what according to our results, can be done using existing monitor data.

## Strengths and limitations

A limitation of this study is the use of interviews that were conducted prior to our study. During the interviews, there were no explicit questions about contextual factors and mechanisms. However, the interview guide to identify key elements was constructed in a way that it was possible to ask questions on why something functioned in a certain way and what supportive and restraining factors were. Another challenge is that the delineation of CMO configurations can be unclear, as it can be difficult to distinguish mechanisms and contexts. To counteract this, the transcripts were coded by two researchers and afterwards experts checked the defined contexts and mechanisms. Moreover, the discussions with the research team (LW, IB, CR) helped to formulate the results and overcome the complexity of what can be seen as a mechanism and what can be seen as a contextual factor. This complexity is also mentioned in other realist evaluation studies [27, 59].

A strength of this study is that contextual factors and mechanisms of Zwolle Healthy City were identified, an exemplary whole system approach that was successful in reducing socioeconomic health inequalities. Moreover, the long duration of Zwolle Healthy City (since 2010) made it possible to describe mechanisms and contextual factors related to the long term implementation and embedding of the approach. Another strength is that it was possible to

interview professionals on all three levels (strategic, tactical and operational) which provided a broad view.

## Implications for practice and research

Since 2010 the integrated community-based approach Zwolle Healthy City has been implemented to stimulate healthy behavior in the municipality of Zwolle. The identified mechanisms and contextual factors based on experiences and perspectives of different stakeholders between 2010 and 2018 in this study may inspire or help other municipalities and organizations with the implementation of an integrated approach. However, not all results are applicable to other local situations: our study showed an influence of the specific local context of Zwolle on the mechanisms and outcomes. This argues for explicitly describing contextual factors of local situations, as we did, so that others can see how relevant the results are for other local situations. The repository from this study can be used as a guide and a source of inspiration for professionals who are also working on implementing an integrated public health approach.

In this study a realist evaluation perspective was used which was suitable for looking back on the implementation of Zwolle Healthy City. A recommendation for future research would be to study whether the mechanisms and contextual factors are also present in other integrated public health approaches.

## Conclusion

This study showed how mechanisms and contextual factors are associated with the key elements of the success of Zwolle Healthy City, an integrated community-based approach aimed at reducing socioeconomic health inequalities. Zwolle Healthy City is a complex multilevel and multisector approach, and using a realist evaluation perspective made it possible to show the critical factors and complexity in a structured manner. The CMO configurations (context + mechanism = outcome) provide insights for professionals who work on implementing an integrated approach for reducing socioeconomic health inequalities, or similar.

## Supporting information

**S1 File. Repository.** CMO configurations of the Zwolle Healthy City approach.
(DOCX)

**S2 File. RAMESES II list of items to be included when reporting realist evaluations.**
(DOCX)

**S3 File. Infographic.** Mechanisms and contextual factors related to key elements of Zwolle Healthy City.
(PDF)

## Author Contributions

**Formal analysis:** Lisa Wilderink, Annemijn Visscher.

**Funding acquisition:** Ingrid Bakker, Carry M. Renders.

**Supervision:** Ingrid Bakker, Albertine J. Schuit, Jacob C. Seidell, Carry M. Renders.

**Writing – original draft:** Lisa Wilderink, Annemijn Visscher.

**Writing – review & editing:** Ingrid Bakker, Albertine J. Schuit, Jacob C. Seidell, Carry M. Renders.

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
