## [Decision Letter · Decision Letter 0]

29 Jun 2022

PONE-D-21-32709Mechanisms and contextual factors related to key elements of a successful integrated community-based approach aimed at reducing socioeconomic health inequalities in the Netherlands: A realist evaluation.PLOS ONE

Dear Dr. Wilderink,

Thank you for submitting your manuscript to PLOS ONE. After careful consideration, we feel that it has merit but does not fully meet PLOS ONE’s publication criteria as it currently stands. Therefore, we invite you to submit a revised version of the manuscript that addresses the points raised during the review process.

Specifically, we require that experiments, statistics, and other analyses are performed to a high technical standard and are described in sufficient detail. Both reviewer raised significant concerns regarding your methodology. Please have those comments addressed point-by-point.

We look forward to receiving your revised manuscript.

Kind regards,

Jianhong Zhou

Staff Editor

PLOS ONE

Journal Requirements:

Reviewers' comments:

Reviewer's Responses to Questions

**Comments to the Author**

1. Is the manuscript technically sound, and do the data support the conclusions?

Reviewer #1: No

Reviewer #2: Partly

2. Has the statistical analysis been performed appropriately and rigorously? 

Reviewer #1: N/A

Reviewer #2: N/A

3. Have the authors made all data underlying the findings in their manuscript fully available?

Reviewer #1: Yes

Reviewer #2: Yes

4. Is the manuscript presented in an intelligible fashion and written in standard English?

Reviewer #1: Yes

Reviewer #2: Yes

5. Review Comments to the Author

Reviewer #1: Dear Authors,

Thank you for the submission. While your paper is very interesting and well written, it is methodologically flawed and for this reason needs complete revision and reanalysis.

Firstly, it is not a realist evaluation. Secondly, while you attempted to apply realist data analysis techniques, this does not appear to be done in line with realist principles of generative causation and retroduction.

Best of luck with the rest of the journey!

Reviewer #2: Overall this is an interesting paper, but there are a number of methodological issues in the manuscript:

- You state (line 101 and elsewhere) that you used realist synthesis to analyse qualitative data. This may be confusing to people less familiar with realist thinking, who see realist synthesis as very distinct from evaluation and focused on the synthesis of literature rather than the analysis of primary data. I would simply say that you have applied a realist logic of analysis to the qualitative data collected.

- Your definition of mechanisms is erroneous. Mechanisms are not the 'things' introduced by an intervention, but the responses that respondents might have to those, in context. They are the invisible forces that drive change, such as fear, sense of belonging etc. Some of the literature describe mechanisms as the combination of resources introduced by an intervention, and the reasoning of actors as a result of this introduction. Others place the resource within the context. Either way, the element that distinguishes mechanisms from other constructs is that they are intangible and linked to the reactions and thinking of actors and systems. You do however talk about these mechanisms, in their true sense, in the text, but the cross references (Mx.x) are beside the tangible aspects of the intervention.

- You present the findings per outcome category. Perhaps this could have been mapped out a bit better, if you had given an overview of all outcomes at each level of the system to situate the reader better. However more problematically, some of the text under outcomes contain only mechanisms. Without understanding of the context that facilitated the triggering of that M, it is difficult to distil the originality of the data. Disaggregating C from M does not make a lot of sense in realism. You make that point very well in the section on consulting experts - any one context can be both facilitative or constraining; it depends on the mechanism and therefore it is counter intuitive to separate them.

- In realist evaluation, the point of the CMO configuration is that a specific M interacts with a specific C to lead to a specific O. It is difficult to see how this is applied when you have 6 Ms and 7 Cs to one O. I think your overall logic of analysis is realist, but it is not expressed in a way that is consistent with realist thinking.

I do like the fact that you are presenting your findings at three levels of the system; this is entirely coherent with realist thinking. The discussion is strong and well written.

6. PLOS authors have the option to publish the peer review history of their article (what does this mean?). If published, this will include your full peer review and any attached files.

Reviewer #1: No

Reviewer #2: No

---

## [Author Response · Author response to Decision Letter 0]

5 Aug 2022

Response to reviewers

Reviewer #1 

1. Dear Authors, 

Thank you for the submission. While your paper is very interesting and well written, it is methodologically flawed and for this reason needs complete revision and reanalysis.

Firstly, it is not a realist evaluation. Secondly, while you attempted to apply realist data analysis techniques, this does not appear to be done in line with realist principles of generative causation and retroduction.

Best of luck with the rest of the journey!

Dear reviewer #1,

Thank you for taking the time to read and review our manuscript. Considering your and reviewer 2’s feedback, we agree that our study is indeed not a strict realist evaluation study, and that using the term ‘realist synthesis’ was not correct. We used however a realist evaluation perspective in the analysis of our qualitative data. Our aim was to answer questions as ‘how does the intervention work’ and ‘in what context does it work’, instead of the question ‘does it work’ which is usually answered in a normal evaluation. These are elements of a realist evaluation. Using such a realist perspective helped us in answering these questions by identifying mechanisms and contextual factors in the primary qualitive data. 

We made this more clear in the text by deleting ‘realist evaluation’ from the title of the paper and replacing phrases in the text as ‘using realist evaluation’ for ‘using elements of a realist evaluation perspective’. We also made more clear in the text that we used primary qualitative data.

Moreover, we agree that the realist principle of retroduction and generative causation could be worked out better in the description of mechanisms in the repository and we have made changes accordingly. Mechanisms are not actions or activities themselves, but the underlying processes that made that these actions or activities lead in a certain context to a certain outcome. It’s about the response of the persons involved. Following realist evaluation principles, these actions or activities produce one or more psychological, attitudinal, and behavioral changes. 

Therefore, the answers to the questions ‘how’ and ‘why’ these processes happen and the responses of involved people to these mechanisms are now made more clear in the description of mechanisms in the repository and in the main text.

Reviewer #2 

1. Overall this is an interesting paper, but there are a number of methodological issues in the manuscript:

Dear reviewer #2,

Firstly, thank you for taking the time to read our manuscript and providing feedback.

2. You state (line 101 and elsewhere) that you used realist synthesis to analyse qualitative data. This may be confusing to people less familiar with realist thinking, who see realist synthesis as very distinct from evaluation and focused on the synthesis of literature rather than the analysis of primary data. I would simply say that you have applied a realist logic of analysis to the qualitative data collected. 

Considering your and reviewer 1’s feedback, we indeed believe that using the term realist synthesis is not suitable. We replaced phrases in the text as ‘using realist evaluation’ for ‘using a realist evaluation perspective’ and ‘realist synthesis’ for ‘qualitative data-analysis with a realist perspective’, because we believe this is more appropriate. We made clearer in the text that we used primary qualitative data. Also, we deleted ‘realist evaluation’ from the title of the paper.

Realist evaluation as an approach is method-neutral as both quantitative as qualitative data can be collected. Identifying observed contextual factors and mechanisms can be done by coding qualitive data.

3. Your definition of mechanisms is erroneous. Mechanisms are not the 'things' introduced by an intervention, but the responses that respondents might have to those, in context. They are the invisible forces that drive change, such as fear, sense of belonging etc. Some of the literature describe mechanisms as the combination of resources introduced by an intervention, and the reasoning of actors as a result of this introduction. Others place the resource within the context. Either way, the element that distinguishes mechanisms from other constructs is that they are intangible and linked to the reactions and thinking of actors and systems. You do however talk about these mechanisms, in their true sense, in the text, but the cross references (Mx.x) are beside the tangible aspects of the intervention. 

We changed the operational description of what a mechanism is in table 1, and rephrased the mechanisms in the repository in a way that according to us is still in line with the analyzed data and the correct definition that you describe. We believe that the reactions and thinking of actors was already present in the elaboration on the mechanisms in the repository, but not in the sentence that describes the mechanisms itself. We now changed these sentences in a way that is more in line with your feedback and realist principles. We also did this in the main text in the result section. 

In additions to the ‘things’ introduced by the intervention we added to the mechanisms in the repository how this led to reactions and response of involved professionals and how this in combination with the context led to the described outcome. 

4. You present the findings per outcome category. Perhaps this could have been mapped out a bit better, if you had given an overview of all outcomes at each level of the system to situate the reader better. However more problematically, some of the text under outcomes contain only mechanisms. Without understanding of the context that facilitated the triggering of that M, it is difficult to distil the originality of the data. Disaggregating C from M does not make a lot of sense in realism. You make that point very well in the section on consulting experts - any one context can be both facilitative or constraining; it depends on the mechanism and therefore it is counter intuitive to separate them.

We indeed chose to present the findings per outcome category. We did not choose to provide an overview of the outcomes for every level, because some outcomes are relevant for multiple levels. However, we made more clear by underlining in the description for what level every specific mechanisms is relevant. We hope this helps in guiding the reader better. 

In the original manuscript three mechanisms were described without context (M2.7, M4.2 and M5.2). We understand that only the combination of C, M and O is relevant in realist evaluation. In these three specific cases, no contextual factors were explicitly mentioned by the respondents. We as authors however do believe that these mechanisms should be displayed, because they occurred in the policy context in which the program Zwolle Health City was implemented. Therefore, to meet your feedback we added ‘local policy aims to reduce socioeconomic health inequalities’ as contextual factors to these three mechanisms, because these mechanisms were triggered by the context of implementation of the Zwolle Healthy City approach.

5. In realist evaluation, the point of the CMO configuration is that a specific M interacts with a specific C to lead to a specific O. It is difficult to see how this is applied when you have 6 Ms and 7 Cs to one O. I think your overall logic of analysis is realist, but it is not expressed in a way that is consistent with realist thinking.

We agree that our study is indeed not a strict realist evaluation study. As you mention, we used indeed the realist logic in the analysis of qualitative data. Our aim was to answer questions as ‘how does the intervention work’ and ‘in what context does it work’, instead of the question ‘does it work’ which is commonly answered in regular evaluation studies. This is in line with realist evaluation. Using a realist perspective helped us in answering these questions by identifying mechanisms and contextual factors in the primary qualitive data. 

For this reason, we replaced phrases in the text as ‘using realist evaluation’ for ‘using a realist evaluation perspective’ and ‘realist synthesis’ for ‘qualitative data-analysis with a realist perspective’, because we believe this is more suitable. Also, we deleted ‘realist evaluation’ from the title of the paper.

6. I do like the fact that you are presenting your findings at three levels of the system; this is entirely coherent with realist thinking. The discussion is strong and well written. 

Thank you.

---

## [Decision Letter · Decision Letter 1]

12 Oct 2022

PONE-D-21-32709R1Mechanisms and contextual factors related to key elements of a successful integrated community-based approach aimed at reducing socioeconomic health inequalities in the NetherlandsPLOS ONE

Dear Dr. Wilderink,

Thank you for submitting your manuscript to PLOS ONE. After careful consideration, we feel that it has merit but does not fully meet PLOS ONE’s publication criteria as it currently stands. Therefore, we invite you to submit a revised version of the manuscript that addresses the points raised during the review process.Please note that without addressing the methodological concerns raised by the reviewer(s), we may be unable to  further consider this work for publication. 

We look forward to receiving your revised manuscript.

Kind regards,

Bettye A. Apenteng

Academic Editor

PLOS ONE

Reviewers' comments:

Reviewer's Responses to Questions

**Comments to the Author**

1. If the authors have adequately addressed your comments raised in a previous round of review and you feel that this manuscript is now acceptable for publication, you may indicate that here to bypass the “Comments to the Author” section, enter your conflict of interest statement in the “Confidential to Editor” section, and submit your "Accept" recommendation.

Reviewer #1: (No Response)

2. Is the manuscript technically sound, and do the data support the conclusions?

Reviewer #1: No

3. Has the statistical analysis been performed appropriately and rigorously? 

Reviewer #1: N/A

4. Have the authors made all data underlying the findings in their manuscript fully available?

Reviewer #1: Yes

5. Is the manuscript presented in an intelligible fashion and written in standard English?

Reviewer #1: Yes

6. Review Comments to the Author

Reviewer #1: Dear Authors,

Again thank you for the opportunity to review your paper, again on a very important topic. However, there are still numerous methodological flaws (or, an understanding of the realist perspective) that are present. Reviewing your changes, there were very minimal changes to any content in regards to the methodology, only wording which rephrased realist evaluation to realist evaluation logic or perspective. However, it is unclear how you followed realist evaluation logic or a perspective. Realist evaluation logic at minimum would, I believe, include some component of theory generation and refinement.

You possibly applied realist lens to your analysis. However, this too was not always clear and appears to miss core components of generative causation and retroduction, which are two necessary components of any realist analysis.

I cannot, therefore, provide a more detailed review of the paper until these core methodological components are expanded and addressed. Please could you, therefore:

• Clearly define how this study incorporated realist evaluation logic

• Please clearly describe your analysis process, and how exactly this took a realist perspective. Identifying Contexts and Mechanisms alone is not sufficient to be a realist ‘study’. Generative causation is fundamental, and this was not very clear in your paper. Notably, more information on the relationship between the Cs Ms and Os needs to be given, and what these specific elements contribute.

• Please review RAMESES II guidelines to get further input to inform your write-up.

I very much appreciate the focus on understanding ‘how does it work’, not just if it works. There are however other approaches to answer this than a realist approach, which again is a clear methodology which requires strong considerations of theory development/refinement, generative causation and retroduction.

7. PLOS authors have the option to publish the peer review history of their article (what does this mean?). If published, this will include your full peer review and any attached files.

Reviewer #1: No

---

## [Author Response · Author response to Decision Letter 1]

13 Jan 2023

Rebuttal 2

Reviewer #1: 

Dear Authors,

Again thank you for the opportunity to review your paper, again on a very important topic. However, there are still numerous methodological flaws (or, an understanding of the realist perspective) that are present. Reviewing your changes, there were very minimal changes to any content in regards to the methodology, only wording which rephrased realist evaluation to realist evaluation logic or perspective. However, it is unclear how you followed realist evaluation logic or a perspective. Realist evaluation logic at minimum would, I believe, include some component of theory generation and refinement.

You possibly applied realist lens to your analysis. However, this too was not always clear and appears to miss core components of generative causation and retroduction, which are two necessary components of any realist analysis.

I cannot, therefore, provide a more detailed review of the paper until these core methodological components are expanded and addressed. Please could you, therefore:

Suggestion reviewer Response authors

Clearly define how this study incorporated realist evaluation logic In the introduction as well in the methods section we now describe how the logic model is seen as the initial programme theory of the Zwolle Healthy City approach and this realist evaluation study contributes to theory refinement by using the stakeholder perspective. Stakeholders’ ideas on what mechanisms and contextual factors led to certain outcomes is a refinement and deepening of this initial programme theory.

Also in the discussion section we now describe that this study showed a further (practical) translation of the initial programme theory. We describe in what way this deepening based on stakeholders’ perspectives led to additions and adjustments of the initial programme theory. 

Please clearly describe your analysis process, and how exactly this took a realist perspective. Identifying Contexts and Mechanisms alone is not sufficient to be a realist ‘study’. Generative causation is fundamental, and this was not very clear in your paper. In the methods section we now added how we took a realist perspective in the data analysis by seeking certain phrases that tell us more about the generative causation relations between C, M and O’s. 

Notably, more information on the relationship between the Cs Ms and Os needs to be given, and what these specific elements contribute. We agree that the relationship between the Cs Ms and Os was not made clear enough in the previous version. This was because we decided to disentangle the M’s and C’s for every O, and describe them separately. In the results section of this revised version, we describe how the combination of the Cs and Ms led to the Os.

Please review RAMESES II guidelines to get further input to inform your write-up. Thank you for the suggestion. An overview of whether and how the different items of the RAMESES II guideline are included in our paper is now added as a supplementary file.

I very much appreciate the focus on understanding ‘how does it work’, not just if it works. There are however other approaches to answer this than a realist approach, which again is a clear methodology which requires strong considerations of theory development/refinement, generative causation and retroduction.

---

## [Decision Letter · Decision Letter 2]

13 Feb 2023

PONE-D-21-32709R2Mechanisms and contextual factors related to key elements of a successful integrated community-based approach aimed at reducing socioeconomic health inequalities in the Netherlands: a realist evaluation perspectivePLOS ONE

Dear Dr. Wilderink,

Thank you for submitting your manuscript to PLOS ONE. After careful consideration, we feel that it has merit but does not fully meet PLOS ONE’s publication criteria as it currently stands. Therefore, we invite you to submit a revised version of the manuscript that addresses the points raised during the review process. The reviewer(s) recommend some additional changes to the manuscript to improve its rigor. The reviewer(s) comments are enclosed in the attached document. 

We look forward to receiving your revised manuscript.

Kind regards,

Bettye A. Apenteng

Academic Editor

PLOS ONE

Journal Requirements:

Reviewers' comments:

Reviewer's Responses to Questions

**Comments to the Author**

1. If the authors have adequately addressed your comments raised in a previous round of review and you feel that this manuscript is now acceptable for publication, you may indicate that here to bypass the “Comments to the Author” section, enter your conflict of interest statement in the “Confidential to Editor” section, and submit your "Accept" recommendation.

Reviewer #3: (No Response)

2. Is the manuscript technically sound, and do the data support the conclusions?

Reviewer #3: Partly

3. Has the statistical analysis been performed appropriately and rigorously? 

Reviewer #3: I Don't Know

4. Have the authors made all data underlying the findings in their manuscript fully available?

Reviewer #3: Yes

5. Is the manuscript presented in an intelligible fashion and written in standard English?

Reviewer #3: Yes

6. Review Comments to the Author

Reviewer #3: Thank you for the opportunity to review this interesting paper addressing the important topic of methods to tackle socioeconomic health inequalities. It is clear you have spent time making substantial revisions in relation to the methodology, and thank you for including the RAMSES guidelines checklist. However, there are some methodological issues that still need to be resolved. Please see attached table. I do encourage you to persist with this paper.

7. PLOS authors have the option to publish the peer review history of their article (what does this mean?). If published, this will include your full peer review and any attached files.

Reviewer #3: No

---

## [Author Response · Author response to Decision Letter 2]

19 Mar 2023

Reviewer:

Thank you for the opportunity to review your interesting manuscript on the important topic of regarding an approach to addressing socioeconomic health inequalities. Please note this is my first review of your paper, but I have read the previous reviewer’s comments and your responses. 

It is clear you have made some major revisions to address the Realist positioning of this paper, and thank you for including the RAMSES guidelines checklist. However, there are a few methodological matters than remain unclear in your paper:

Authors:

Thank you very much for taking the time to review our paper and providing it with your helpful and detailed feedback. With the following changes, we hope to meet your comments.

Comment reviewer Response author

Abstract 

You may wish to consider what readers will be most interested in when reading the results section of your abstract – I suspect a summary of the two/three most key CMOCs will be more of a hook that a quantitative number of CMOCs returnend. 

We agree, thank you for the suggestion. We now describe two CMOs in the abstract. Because the abstract cannot exceed 300 words, we had to make some other changes to the abstract. 

Introduction/ background 

Thank you for adding a paragraph clarifying that the study intends to refine an existing logic model.

Purpose of the study: why was a Realist approach appropriate? As an earlier reviewer commented – other approaches can be use to address how & why something works – why Realism? It may be useful here to think about why you were interested in context and causation…. Perhaps for those intending to understand how to implement the model elsewhere? 

We now describe why we used realist evaluation: because realism specifically is based on the idea that the same approach will not work the same everywhere and for everyone. In line 111-116 (in the track changes document) we now added why we chose for realism. Not only because Zwolle can continue to adapt based on these insights, but also to implement the approach elsewhere indeed. Our paper shows a way such an approach can be implemented, and is not a ‘blueprint’ of a one size fits all approach that can be implemented in the exact same form. 

Methods 

1. Good practice is to explain why you used the type of data you used & why you invited the participants that you did – how was this best placed to inform the refinement of your programme theory? 

In the discussion/limitations section you mention that the interviews were undertaken prior to the design of this study and that this was a limiting factor in understanding contextual factors in the CMOC. Although the explanation of the limitation is well placed in the discussion, the source of the interviews is perhaps a point for the methods section, particularly in light of the usual expectation to describe why the data were best placed to address your realist programme theory refinement. If these data were collected for a different purpose but have been utilised for this secondary purpose in this realist theory building – please state.

2. Usually the exact number and role of interviewees is the first thing reported in the results section (unless these were secondary data? In which case could be considered part of the ‘setting’). 

The analysis section begins by discussing the use of a Braun & Clarke style thematic analysis – I would question whether this is methodologically congruent with the Realist evaluation approach. As you later mention that the interviews were collected originally for a different purpose I was unsure whether this element took place as part of a different research phase/question.

3. Do check you are using expected Realist terminology and that you are using it correctly throughout. E.g: Retroductive as an inductive/deductive cycle. “demi-regularities” rather than “patterns”.

4. The use of ‘experts’ to agree CMOCs is a fantastic aid to your rigour. However, by placing the changes/additions made by these experts in your results section rather than your methods section I was left confused whether you considered the final CMOCs to be those made before the experts added their thoughts or after. Assuming that the final CMOCs are those made after the input of the experts I would move the entire section to your methods. Additionally, there is (I think) a typo in the section that needs moving: the sentence states that experts missed two CMOCs, but I think the intention is that the CMOCs identified two CMOCs that were previously missed? 

1. We now made more clear in the methods section that we used the original data for this secondary purpose (line 133-139 in the manuscript with track changes).

2. We agree. Because we used secondary data, we removed the part in the data analysis sections where we describe how the original data was analysed. This thematic analysis in Braun & Clarke style took place in a different research phase (the phase where we identified the key elements).

3. Thank you for the suggestions. We changed this in the manuscript.

4. Thank you for the suggestion. The final changes were indeed made after the input of experts. However, we chose to keep the results of this expert meeting at the end of the results section. This is because we do not want to describe in the methods sections how the input of experts changed CMOCs, when the reader did not yet read about those CMOCs. In the methods section we do however describe that we conducted an expert meeting and what we aimed to do in this meeting (line 187-199). We hope that you can agree on our view.

We also corrected the typo you mention (line 430-431).

Results 

There are some detailed CMOCs reported. This demonstrates some thorough analysis and retroductive thinking. 

However, you state your intention was refine your initial programme theory. Therefore, you need to discuss whether these findings confirm/disprove/add detail to/other your initial programme theory. Depending on which, this may be a statement near the beginning of your results, or may be something you address under each outcome. One thing I found a little confusing, for example, was whether the outcomes had changed as part of your theory refinement, or whether these had stayed the same and the context/mechanism configuration causing those outcomes had changed. 

It was our purpose to add more detail to the initial programme theory by defining mechanisms and contextual factors. We now state this at the start of the results section.

Discussion 

The results are placed in the context of wider literature.

Your reflections on initial programme theory might be better placed at the beginning of this section. 

We agree, we moved this paragraph to the beginning of the section.

Throughout 

A few times you mention “specifying” the programme theory. This a little confusing. Are you defining an initial programme theory, or are you refining an existing one? If the latter, consider removing/changing this word 

We are refining an existing one by providing more details. Therefore, we changed the word ‘specifying’ to ‘adding detail to’ throughout the paper.

---

## [Decision Letter · Decision Letter 3]

12 Apr 2023

Mechanisms and contextual factors related to key elements of a successful integrated community-based approach aimed at reducing socioeconomic health inequalities in the Netherlands: a realist evaluation perspective

PONE-D-21-32709R3

Dear Dr. Wilderink,

We’re pleased to inform you that your manuscript has been judged scientifically suitable for publication and will be formally accepted for publication once it meets all outstanding technical requirements.

Kind regards,

Bettye A. Apenteng

Academic Editor

PLOS ONE

Additional Editor Comments (optional):

Reviewers' comments:

Reviewer's Responses to Questions

**Comments to the Author**

1. If the authors have adequately addressed your comments raised in a previous round of review and you feel that this manuscript is now acceptable for publication, you may indicate that here to bypass the “Comments to the Author” section, enter your conflict of interest statement in the “Confidential to Editor” section, and submit your "Accept" recommendation.

Reviewer #3: All comments have been addressed

2. Is the manuscript technically sound, and do the data support the conclusions?

Reviewer #3: Yes

3. Has the statistical analysis been performed appropriately and rigorously? 

Reviewer #3: N/A

4. Have the authors made all data underlying the findings in their manuscript fully available?

Reviewer #3: Yes

5. Is the manuscript presented in an intelligible fashion and written in standard English?

Reviewer #3: Yes

6. Review Comments to the Author

Reviewer #3: The Realist methodology is much more clearly explained, and previous reviewer comments regarding this have been addressed.

7. PLOS authors have the option to publish the peer review history of their article (what does this mean?). If published, this will include your full peer review and any attached files.

Reviewer #3: No

---

## [Editor Report · Acceptance letter]

8 May 2023

PONE-D-21-32709R3 

Mechanisms and contextual factors related to key elements of a successful integrated community-based approach aimed at reducing socioeconomic health inequalities in the Netherlands: a realist evaluation perspective 

Dear Dr. Wilderink:

I'm pleased to inform you that your manuscript has been deemed suitable for publication in PLOS ONE. Congratulations! Your manuscript is now with our production department. 

Kind regards, 

on behalf of

Dr. Bettye A. Apenteng 

Academic Editor

PLOS ONE